# Ultrasound Imaging of Subtalar Joint Instability for Chronic Ankle Instability

**DOI:** 10.3390/healthcare11152227

**Published:** 2023-08-07

**Authors:** Shintarou Kudo, Tsutomu Aoyagi, Takumi Kobayashi, Yuta Koshino, Mutsuaki Edama

**Affiliations:** 1Department of Physical Therapy, Morinomiya University of Medical Sciences, Osaka 559-8611, Japan; aoyagi-tsutomu@ar-ex.jp; 2Department of the Rehabilitation, Oyamadai Orthopedics Clinic Tokyo Arthroscopy Center, Tokyo 158-0082, Japan; 3Department of Rehabilitation, Hokkaido Chitose College of Rehabilitation, Chitose 066-0055, Japan; kobatakku@gmail.com; 4Faculty of Health Sciences, Hokkaido University, Sapporo 060-0812, Japan; y-t-1-6@hs.hokudai.ac.jp; 5Institute for Human Movement and Medical Sciences, Niigata University of Health and Welfare, Niigata 950-3102, Japan; edama@nuhw.ac.jp

**Keywords:** chronic ankle instability, subtalar joint, ultrasonography

## Abstract

The purpose of this study was to develop the assessment of subtalar joint instability with chronic ankle instability (CAI) using ultrasonography. Forty-six patients with anterior talofibular ligament (ATFL) abnormalities and a history of ankle sprain were divided into CAI (21.2 ± 5.9 y/o, 7 males and 17 females) and asymptomatic groups (21.0 ± 7.4 y/o, 9 males and 12 females) on the basis of subjective ankle instability assessed using the CAIT and the Ankle Instability Instrument Tool (AIIT). Twenty-six age-matched feet participated in a control group (18.9 ± 7.0 y/o, 9 males and 17 females). Ultrasound measurements of the width of the posterior subtalar joint facet were obtained at rest and maximum ankle inversion (subtalar joint excursion; STJE). The differences in STJE among the three groups were assessed by one-way ANOVA. The relationship between STJE and subjective ankle instability was assessed using Spearman’s correlation tests. The STJE value was significantly greater in the CAI group (2.3 ± 0.8 mm) than in the asymptomatic (1.0 ±0.4 mm) and control groups (0.8 ±0.2 mm) (*p* < 0.001, effect size: 0.64). STJE had significant negative correlations with CAIT (r = −0.71, *p* < 0.01), and significant positive correlations with AIIT (r = 0.74, *p* < 0.01). The cut-off value to distinguish between the CAI and asymptomatic groups was 1.7 mm using the ROC curve.

## 1. Introduction

Chronic ankle instability (CAI) due to repeated ankle sprains causes various dysfunctions [1] and decreases patients’ quality of life [2]. In subjects with CAI, injury of the lateral ankle ligaments, such as the anterior talofibular ligament (ATFL) or the calcaneofibular ligament (CFL), causes mechanical instability of the talocrural joint, and ligament reconstruction is performed in cases resistant to conservative treatment [3]. Some researchers reported that about 10–25% of CAI patients have damaged ligaments related to the instability of the STJ, such as the interosseous talocalcaneal ligament (ITCL) and the cervical ligament (CL) [4]. Patients with ITCL and CL injuries are significantly more likely to complain of the ankle giving way or pain [5], and instability of the STJ may be involved in the pathophysiology of CAI.

Stress radiography or MRI is mainly used for the diagnosis of STJ instability [3]. The STJ instability has been assessed using the Brodén stress view, which is the angle measured between the talus and calcaneus during maximum inversion of the foot [6]. However, in a recent systematic review, it was concluded that the Brodén stress view does not have sufficient evidence to support its use for the assessment of subtalar instability [7]. In addition, although qualitative assessment of ligaments is possible with MRI [8], quantitative assessment of instability has been difficult [9]. Therefore, quantitative assessment of STJ instability remains inadequate.

Ultrasonography (US) is an imaging modality that can assess joint dynamics non-invasively and in real time. In recent years, assessment of joint instability by ultrasonography has been established for some joints [10,11]. For the ankle joint, a method was proposed to assess the instability of the talocrural joint by observation of the ATFL using US, and high sensitivity and specificity were reported [12]. The CFL is the primary stabilizer of the STJ. However, US does not have high sensitivity for the diagnosis of the CFL. Cheng et al. reported low sensitivity for complete CFL tears (50%) [13]. This may be because of the CFL’s anatomy. It has a concave course, and the tip of the lateral malleolus is obviously protuberant, which makes evaluation of its malleolar insertion more difficult [13]. Moreover, Edama et al. [14] showed that the morphology of the CFL has high variability and affects its function of restricting inversion. Thus, US of the CFL cannot be established as a clinically useful tool. Meanwhile, there have been very few reports on assessing STJ instability by ultrasonography [15], and no method has been established. In addition, we have tried to assess the stability of the STJ by measuring the STJ width. Then, we hypothesized that subjects with CAI had a large amount of excursion of the STJ width between resting positing and maximum inversion position. The purpose of this study was to develop ultrasound assessment of STJ instability in subjects with CAI.

## 2. Materials and Methods

### 2.1. Participants

The study was approved by the Medical Ethics Committee of the authors’ institution (2016–072), and all the subjects provided informed consent for study participation. Forty-six feet of forty-six patients were included in this study. All patients had been diagnosed with a lateral ankle sprain and had undergone physical therapy at one orthopedic clinic from April 2018 to August 2019. The inclusion criteria were a past medical history of several ankle sprains and more than 1 month had passed since the most recent lateral ankle sprain at the time of testing. The exclusion criteria were having tenderness at the ATFL and/or motion pain at maximum inversion stress. Cases with tenderness in the ATFL and/ or motion pain at maximum inversion stress were excluded as there may be some residual abnormality, such as chronic inflammation of the ATFL.

The structure of the ATFL was assessed using US [16], and subjective ankle instability was assessed using two questionnaires that could be used to assess subjective ankle instability, the Cumberland Ankle Instability Tool (CAIT) [17,18] and the Ankle Instability Instrument Tool (AIIT) [19]. Patients who had both less than 24 points on the CAIT and more than five items on the AIIT were diagnosed as having subjective ankle instability. All patients with an abnormal ATFL were divided into the asymptomatic and CAI groups. Patients who had an abnormal ATFL, but did not have subjective ankle instability, were included in the asymptomatic group, whereas patients who had an abnormal ATFL and subjective ankle instability were included in the CAI group. In addition, an age-matched control group (26 feet of 26 persons) with no pain and past medical history of ankle sprain and fracture was selected (Figure 1). Written informed consent was obtained from all subjects.

### 2.2. Procedures

US was performed using B mode with the EUB-7500 linear probe (Hitachi, Ltd., Tokyo, Japan). Assessment of the ATFL by US was performed according to the study of Hua et al. [16] ATFL injury was classified as: (1) ligament tear, a partial or total interruption of the ligament fibers at the fibular side, talar side, or in mid-stance; (2) lax ligament, the ligament remained curved when the ankle was in maximum inversion and plantar flexion; (3) thick ligament, the width of the ligament was >24 mm or >20% of the contralateral normal ligament; (4) ligament absorbed, no ligament fibers were seen; and (5) non-union of an avulsion fracture of the lateral malleolus. Subjects with an abnormal ATFL were divided into a CAI and an asymptomatic group.

The STJ was examined at rest on a bed with approximately 20 degrees of ankle plantarflexion. The linear probe was placed at the antero-inferior aspect of the lateral malleolus. The proximal side of the probe was then attached, and its distal side was rotated backward 30 degrees to capture the STJ (Figure 2). The posterior facet of the STJ was identified with a short axis image of the peroneus tendon drawn on the calcaneal side, and the linear hyperechoic position of the talus and calcaneal bone surface was defined as the edge of the STJ (Figure 3). The STJ width, which was the distance between the edges of STJ, was measured in the rest position and the manually maximum inversion stress position using the caliper function of the US device (measurement accuracy 0.1 mm). The probe was fixed manually, and the probe position was coordinated in each position to obtain a clear image with minimized contact strength. The STJ excursion (STJE) was defined as the difference in STJ width between the rest and inversion position (Figure 2). A single examiner performed all US assessments. The STJE was measured three times, and the mean value was analyzed before both the CAIT and AIIT were answered.

### 2.3. Reliability Assessment

Nine feet of nine subjects (seven males, two females) who did not have any ankle instability participated in this study to examine the reliability of the assessment of STJE. Their mean age was 26.0 ± 2.2 years, height was 165.0 ± 6.1 cm, and weight was 58.2 ± 7.4 kg. Subjects with a past medical history of a sprained ankle within the past month or a past medical history of ankle surgery were excluded. The ultrasound measurements were performed by two testers, one of whom was experienced in US assessment and the other a novice assessor. The experienced tester had 5 years of experience in assessing muscles using ultrasound imaging. Training of the novice rater included basic education in ultrasound imaging techniques and practicing capturing ultrasound images under the observation of the experienced operator with instruction in the measurement technique for 15–30 min/day for 1 month. The two examiners performed assessments three times separately on the same day as the first session, and one examiner performed repeat evaluations again three times on another day as the second session. For examination of intra-rater reliability, ICC (1, k) and the standard error of the mean (SEM) were calculated from the mean value of the first and second sessions. For inter-rater reliability, ICC (2, k) and the SEM were calculated from the mean value of the first session of the two raters. Minimum detectable change 95 (MDC_95_) was calculated from Equation (1).
(1)$${MDC}_{95}=SEM\times 1.96\times \sqrt{2\;}$$

### 2.4. Statistical Analysis

One-way analysis of variance (ANOVA) was used to compare the age, height, and weight of the three groups. Sex and affected side distribution were compared using the chi-squared test. The STJE, the CAIT, and AII were compared among the three groups by ANOVA with Tukey’s test as a post hoc test. Spearman’s correlation tests were performed, with calculation of the correlation coefficients of STJ rest and inversion and STJE with subjective ankle instability based on the CAIT and the AIIT.

The diagnostic performance of the STJE was determined using receiver operating characteristic (ROC) curves. Putative cut-off values for STJE to calculate sensitivity and specificity were calculated. The Youden index was used to determine the best cut-off value. The STJE value corresponding to the highest Youden index was used to divide the patients into asymptomatic and CAI subgroups. All statistical tests were performed using SPSS statistics, version 25.0 (IBM Corp., Armonk, NY, USA), with *p* < 0.05 considered significant.

## 3. Results

### 3.1. Differences in General Characteristics among the Three Groups

The CAI, asymptomatic, and control groups consisted of 25 persons, 21 persons, and 26 persons, respectively. There were no significant differences among the three groups in age, height, weight, sex, and affected side. CAIT were significant differences among the three groups (control group: 29.4 ± 5.7 points, asymptomatic group: 24.9 ± 4.3 points, and CAI group: 14.1 ± 2.0 points). AIIT were significant differences among the three groups (control group: 0.1 ± 1.0 points, asymptomatic group: 3.2 ± 0.9 points, and CAI group: 6.3 ± 0.4 points) (Table 1).

### 3.2. Reliability

All results demonstrated high reliability. Intra-rater reliability within the same day was 0.85 (0.64–0.96). Inter-rater reliability was 0.95 (0.81–0.98). The ICC, SEM, and MDC_95_ are shown in Table 2.

### 3.3. Comparisons among the Three Groups

The outcomes of US imaging of the STJ are shown in Table 3. There was a significant difference in STJ rest among the three groups (Control: 2.1 ± 0.8 mm, asymptomatic: 2.5 ± 0.5 mm, CAI: 2.4 ± 0.6 mm, effect size: 0.03), but there were no significant differences between any of the groups on post hoc comparisons (*p*-value of Control-CAI: 0.06, Control-asymptomatic: 0.13, asymptomatic-CAI: 0.90). STJ inversion showed a significant difference among the groups (Control: 2.9 ± 0.9 mm, asymptomatic: 3.5 ± 0.7 mm, CAI: 3.5 ± 0.7 mm, effect size: 0.50). The STJE value was significantly greater in the CAI group (2.3 ± 0.8 mm) than in the asymptomatic group (1.0 ± 0.4 mm) and control group (0.8 ±0.2 mm) (*p* < 0.001, effect size: 0.64).

### 3.4. Correlation between STJE and Subjective Ankle Instability

Correlation coefficient values are shown in Table 4. STJ rest had no correlations with CAIT (r = 0.12) and AIIT (r = −0.08). Both STJ inversion and STJE had significant negative correlations with CAIT (STJ inversion: r = −0.6, *p* < 0.01, STJE: r = −0.71, *p* < 0.01), and significant positive correlations with AIIT (STJ inversion: r = 0.67, *p* < 0.01, STJE: r = 0.74, *p* < 0.01).

### 3.5. Cut-Off Value

The ROC curve used to examine the diagnostic performance of STJE as a predictor of CAI had an area under the curve of 0.94 (95% CI 0.87–1.00). The highest Youden index was 0.79, which corresponded to a cut-off value of 1.7 mm (Figure 4).

## 4. Discussion

The purpose of this study was to develop ultrasound assessment of the STJ instability in subjects with CAI. The STJ width in inversion and STJE are different among the three groups. The difference in STJ width in inversion in the CAI group with other groups was greater than the MDC_95_ value of STJ inversion, but the SEM and MDC_95_ values of STJ inversion were larger than those of STJE. Moreover, the correlation coefficient value of STJE and subjective ankle instability was stronger than that of the STJ width in inversion. Additionally, the cut-off value to distinguish between the CAI and asymptomatic groups was 1.7 mm. Therefore, increased STJE was one of the important factors related to subjective ankle instability.

In the previous study, STJ assessment did not satisfy the need for a gold standard. Thus, the relationship between the STJ instability and the pathophysiology of CAI had not been clear. Renstrom reported [20] that STJ instability and CAI present very similarly, in that patients suffer from “giving way” in the setting of recurrent sprains. In one study by Meyer et al. [21], 40 patients with acute lateral ankle sprains underwent subtalar arthrograms, and 32 patients (80%) had injured capsuloligamentous structures of both the subtalar and the talocrural joints. In addition, Brantigan et al. estimated [22] that the STJ instability was present in about 10% of patients with lateral ankle ligament instability. Thus, the STJ instability could be related to the pathophysiology of CAI, but the incidence of these injuries was unknown, and there might be a potentially higher incidence of patients with the STJ instability than patients who could be assessed for the STJ instability. In this study, the assessment using US is highly reliable and was clarified in the control, asymptomatic, and CAI groups; moreover, the cut-off value to distinguish between the CAI and asymptomatic groups was 1.7 mm. Thus, we found that subjects with CAI who had subjective ankle instability had a larger excursion of the STJ.

Injuries of the ligaments of the STJ most often occur in conjunction with a lateral ankle sprain [20]. The excessive inversion of the ankle and internal rotation of the talus were movements that led to a lateral ankle sprain [23]. The STJ has extensive amounts of ligamentous structures, such as the ITCL, CL, CFL, and the lateral root of the inferior extensor retinaculum [20,24]. The ITCL and CL have functions to stabilize talus rotation [25]. The talocrural joint is called the “mortise” joint and is formed by the articulation of the dome of the talus, the medial malleolus, the tibial plafond, and the lateral malleolus. The talocrural joint might be thought of as a hinge joint that allowed the motions of plantarflexion and dorsiflexion. The isolated movement of the talocrural joint was primarily in the sagittal plane, but small amounts of transverse- and frontal-plane motion also occurred about the oblique axis of rotation [26]. Inversion and internal rotation of the ankle consisted of a large amount of movement in the STJ. Therefore, the STJ was sometimes injured in inversion and internal rotation of the talus with a lateral ankle sprain. The STJE, which was calculated from the difference in the STJ width in rest and maximum inversion position, could be related to the inversion excursion of the STJ. Therefore, the subjects with CAI, with the possibility of STJ instability, showed higher values of STJE.

Sprains of the STJ were difficult to define and were even more difficult to diagnose, because of the unclear assessment of the STJ instability. All subjects of both the CAI and asymptomatic groups had some abnormality of the ATFL and a past medical history of multiple ankle sprains. Thus, CAI patients had a past medical history of multiple ankle sprains, an abnormal ATFL, and STJE greater than 1.7 mm. We suggest that those ligament structures of patients with greater STJE require detailed assessment by magnetic resonance imaging (MRI). When these diagnostic strategies for STJ instability become established, how much the STJ instability affects the symptoms of ankle instability will be clarified. Recently, research has focused on the sensorimotor alterations in CAI. In assessing the effects of CAI on balance and other performance indicators, it is often discussed that the contralateral limb may perform similarly worse compared to healthy subjects, leading to the theory of altered sensorimotor control in CAI that affects both limbs [27,28]. The subtalar joint mobility restricted the ligament structure such as ITCL and CL in the sinus tarsi, which is the anterior part of the subtalar joint and has a fat pad with the important role of proprioception [29]. If the subtalar joint instability in CAI is the result of an injury to the ligamentum structure of the subtalar joint, it may influence the sensorimotor alteration in CAI. Thus, it is necessary to assess the relationship between instability of the subtalar joint and sensorimotor alteration in CAI in further studies.

This study had some limitations. The STJ movement was classified as inversion-eversion around a single oblique axis inclined an average of 42 degrees in the sagittal plane and medial deviation of 23 degrees in the axial plane related to the longitudinal axis of the foot. In the open kinetic chain, supination consists of plantarflexion, inversion, and internal rotation [30], because the talocrural and STJs each have oblique axes of rotation. In the present study, STJE was assessed in inversion movement, but the other components, such as antero-posterior (AP) translation and internal rotation of the talus, were not assessable in detail.

In the present study, inversion stress was applied manually, and the reliability of STJE was only assessed in normal volunteers. Thus, all patients were assessed by a single examiner, but the degree of manual stress might affect the STJE value.

The criteria of the IAC could not be applied to this study. According to the criteria of the International Ankle Consortium (IAC) [31], CAI is defined as patients with both a history of several ankle sprains and subjective ankle instability who had not had an ankle sprain within 3 months from time of testing. However, many CAI patients were free from ankle pain at 1 or 2 months, and returned to play regardless of the functional limitation such as the subjective ankle instability. Therefore, we could not follow up the patients for more than 3 months in the clinical setting. In this study, the patients with subjective instability and a history of several ankle sprains, without any pain, and more than 1 month after the injury, were included in the CAI group.

The CFL is known to be integral in preventing excessive inversion and external rotation of the STJ [32,33]. CFL function could not be assessed because the assessment of the CFL using US has insufficient reliability and it is difficult to precisely assess the ligament’s function [13]. The function of the CFL and stability of the STJ could be assessed using MRI and arthroscopy. However, those could not be performed for subjects without any ankle pain. Therefore, it was not clear whether the ligament structures stabilizing the STJ, such as the CFL, ITCL, and CL, were injured. Peroneal tendinopathy is sometimes obtained in the CAI patients [34]. The peroneus tendon can be seen by US when capturing the width of the STJ in US. However, we cannot record the number of abnormal peroneus tendons. Therefore, we could not discuss the relationship between abnormality of the peroneus tendon and instability of the STJ.

The most important limitation of this study was that there was no reference test. However, subjects with subjective instability had increased STJE values; therefore, it could be said that they had the STJ instability during inversion of the ankle using STJE values. Further studies are needed to investigate the relationship between the development of CAI and CFL function using MRI.

## 5. Conclusions

The subtalar joint instability is known to be one of the factors affecting CAI; however, there were no methods to assess it in clinical settings. The STJE during inversion of the ankle could be assessed non-invasively by US. Individuals with CAI had increased STJE that was associated with subjective ankle instability. The STJE may be reflected in the subtalar joint instability, therefore it was necessary to assess not only the ATFL but also the STJ excursion in the subject with CAI. However, CAI is affected by multiple factors, and assessment of the CAI is performed in multiple dimensions, which are instability of the ligamentum structure, muscle strength, motor control, and sensory–motor function. In further studies, clarified pathophysiology may be needed to assess between several factors and subtalar joint instability.

## Figures and Tables

**Figure 1 healthcare-11-02227-f001:**
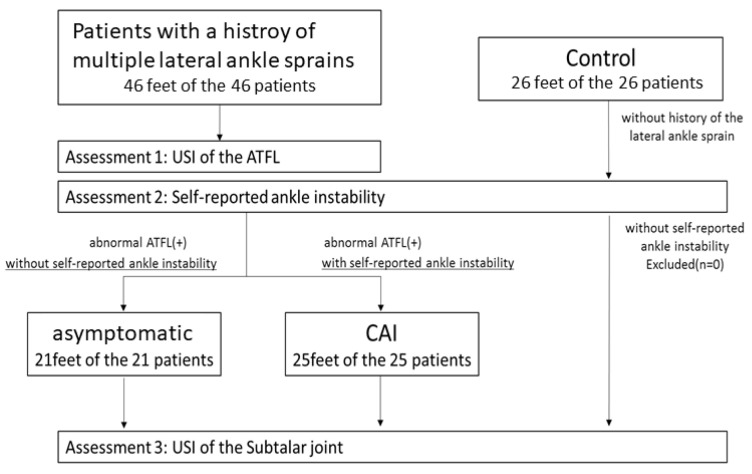
Study flow diagram. CAI: chronic ankle instability, USI: ultrasound imaging, ATFL: anterior talofibular ligament.

**Figure 2 healthcare-11-02227-f002:**
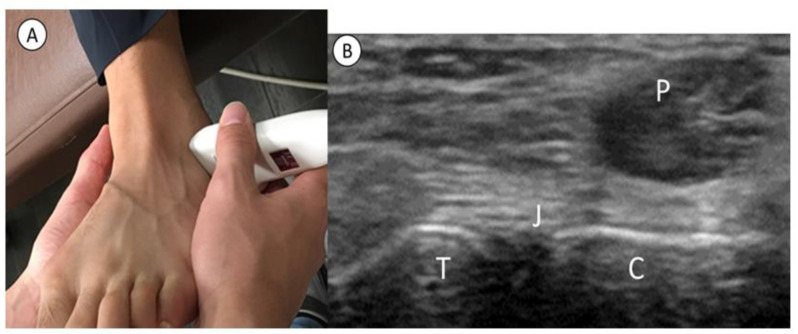
Ultrasonography image of the subtalar joint (**A**) A probe is placed on the antero-inferior aspect of the lateral malleolus. Next, a probe is fixed proximally, and its distal end is slid posteriorly at about 30 degrees to capture the subtalar joint. (**B**) ultrasound image, P: peroneal tendon, T: talus, J: subtalar joint, C: calcaneus.

**Figure 3 healthcare-11-02227-f003:**
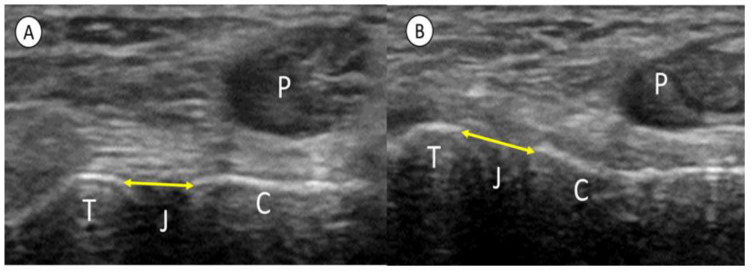
Measurement of the subtalar joint width (STJ width) (**A**) STJ width in a rest (**B**) STJ width in maximum inversion. The width of the talus and calcaneus (arrows) is measured at rest (**A**) and at maximum inversion of the calcaneus (**B**) by electronic calipers.

**Figure 4 healthcare-11-02227-f004:**
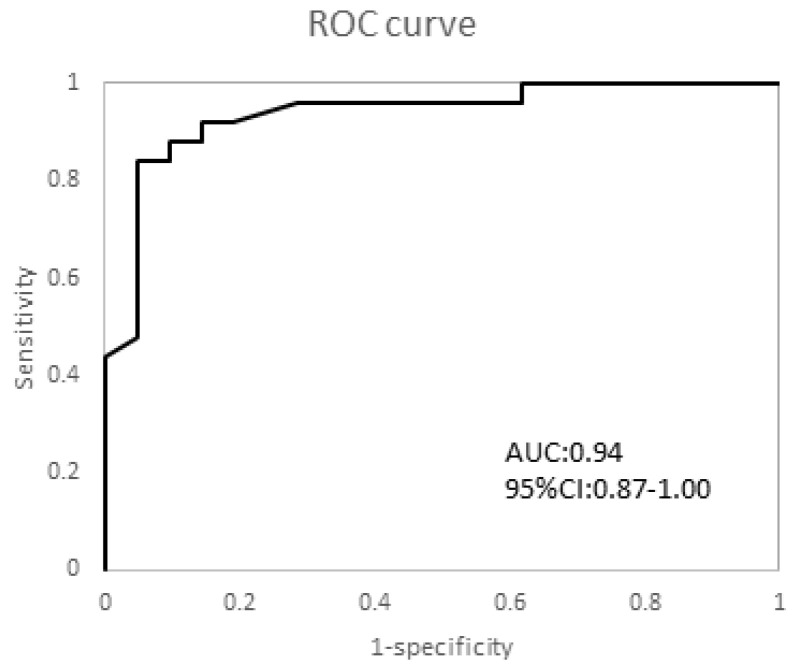
Receiver operating characteristic curve. ROC curve of STJE, with the cut-off value set at 1.7 mm.

**Table 1 healthcare-11-02227-t001:** General characteristics of the three groups.

	Control	Asymptomatic	CAI	*p*-Value
	26 persons	21 persons	25 persons	
Age (y)	18.9 ± 7.0	21.0 ± 7.4	21.2 ± 5.9	0.43
Height (cm)	164.5 ± 7.4	166.7 ± 8.8	164.3 ± 7.8	0.53
Weight (kg)	59.0 ± 8.8	59.7 ± 8.8	56.6 ± 8.8	0.48
Sex M/F	9/17	9/12	7/17	0.62
Side (R/L)	8/18	11/10	13/11	0.17
AIIT	0.1 ± 1.0	3.2 ± 0.9	6.3 ± 0.4	<0.001 *
CAIT	29.4 ± 5.7	24.9 ± 4.3	14.1 ± 2.0	<0.01 *

*: significant difference among the three groups. AIIT: Ankle instability Instrument Tool, CAIT: Cumberland Ankle Instability Tool.

**Table 2 healthcare-11-02227-t002:** Intra- and inter-rater reliabilities of STJE.

	Reliability Type	ICC (95%CI)	SEM	MDC95
STJ rest	Intra-rater	0.95 (0.89–0.98)	0.08 (0.06–0.10)	0.22
	Inter-rater	0.81 (−0.02–0.96)	0.17 (0.12–0.31)	0.47
STJ inversion	Intra-rater	0.87 (0.73–0.96)	0.23 (0.19–0.30)	0.63
	Inter-rater	0.88 (0.31–0.98)	0.25 (0.17–0.46)	0.69
STJE	Intra-rater	0.84 (0.64–0.98)	0.13 (0.11–0.17)	0.36
	Inter-rater	0.95 (0.81–0.98)	0.14 (0.10–0.27)	0.39

STJ: subtalar joint, STJE: the subtalar joint excursion.

**Table 3 healthcare-11-02227-t003:** Differences in STJ instability among the three groups.

	Control	Asymptomatic	CAI	*p*-Value	ES (η^2^)	Post Hoc	
STJ (mm)							
rest	2.1 ± 0.8	2.5 ± 0.5	2.4 ± 0.6	<0.05	0.03	Control-CAI Control-asymptomatic asymptomatic-CAI	0.06 0.13 0.90
inversion	2.9 ± 0.9	3.5 ± 0.7	4.7 ± 0.8	<0.001	0.50	Control-CAI, Control-asymptomatic asymptomatic-CAI	0.01 <0.001 <0.001
STJE	0.8 ± 0.2	1.0 ± 0.4	2.3 ± 0.8	<0.001	0.64	Control-CAI, Control-asymptomatic asymptomatic-CAI	<0.001 0.18 <0.001

Values are means ± standard deviation. STJ: subtalar joint, STJE: the subtalar joint excursion, ES: effect size.

**Table 4 healthcare-11-02227-t004:** Correlation coefficient values between subtalar joint instability and self-reported ankle instability.

	CAIT	AIIT
STJ rest	0.12	−0.08
STJ inversion	−0.60 *	0.67 *
STJE	−0.71 *	0.74 *

Values are correlation coefficient values (*r*). *: *p*<0.01. STJ: subtalar joint, STJE: the subtalar joint excursion.

## Data Availability

The data are available upon request from the corresponding author.

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
