# Peer review of "Ultrasound Imaging of Subtalar Joint Instability for Chronic Ankle Instability"

_healthcare, 2023, doi:10.3390/healthcare11152227_

Round 1

Reviewer 1 Report

- please re-phrase abstract, more concise

- explain why tenderness of ATFL is exclusion criteria, isn´t that also part of insufficiency or abnormal ATFL? 

- improve grammar and quality of speech (e.g. line 77)

- did you look at the peroneals? clinically or by US?

- so subjective and relevant instability only exists if subtalar ligaments are damaged?

- "undergoing" physiotherapy: so all participants in all groups have had physiotherapy? what kind of physio? for how long? any differences in between groups?

I would recommend english editing service

Author Response

Dear reviewer

Thank you for your letter. We are grateful for the detailed feedback provided from you, which we feel has helped us to significantly improve the paper. 

Thank you again for your thoughtful comments, and we look forward to hearing from you soon.

Reviewer 2 Report

I would like to thank the editor for the opportunity to review this interesting manuscript. The use of ultrasonography to detect ankle joint abnormalities in people with CAI can be of great interest to provide better insights both in terms of diagnostics and evaluation, and for better understanding of the possible pathophysiological mechanisms.
I have some suggestions I hope can help to improve the manuscript before acceptance:

Abstract:

- Please, explain Coper group

- Please, provide sex distribution and age of the included participants per group

- Please, provide measures of effect size for comparisons and actual p values.

- Please, also provide p values for correlation

- It is not clear from the abstract how the authors provided the cut-off value (ROC analysis?)

- There are some errors in the sentence (". And [...]")

Discussion:

-I think that the study is well performed and well discussed. I have just one curiosity: this ultrasound evaluation can help to identify morphological alterations associated with CAI. When evaluating CAI effects on balance and other performance indicators, it is often discussed that the contralateral limb might present similar worse outcomes compared to healthy subjects, leading to the theory of an altered sensorimotor control in CAI, affecting both limbs (Deodato et al., Clin Biomech, 2023; Hertel & Corbett, J Athl Train, 2019). I think that based on the authors' data, it might be interesting to discuss the relationship between morphological and sensorimotor alterations in CAI functional evaluation.

Author Response

Thank you for your letter. We are grateful for the detailed feedback provided from you, which we feel has helped us to significantly improve the paper. Attached are our point-by-point responses to the reviewers’ comments and our revised manuscript, which we hope will now meet with your approval. 

Thank you again for your thoughtful comments, and we look forward to hearing from you soon.
